# *In vivo* biomechanical responses of neonatal brachial plexus when subjected to stretch

**Anita Singh** [1]*, **Virginia Orozco** [2], **Sriram Balasubramanian** [2]

1 Bioengineering Department, Temple University, Philadelphia, Pennsylvania, United States of America,
2 School of Biomedical Engineering, Science and Health Systems, Drexel University, Philadelphia, Pennsylvania, United States of America

* anita.singh0001@temple.edu

## Abstract

Neonatal brachial plexus palsy (NBPP) results from over-stretching of the neonatal brachial plexus during complicated birthing scenarios. The lack of information on the biomechanical response of the neonatal brachial plexus complex when subjected to stretch limits our understanding of the NBPP injury mechanism. This study aims to fill that critical gap by using a neonatal piglet animal model and providing the *in vivo* biomechanical properties of the neonatal brachial plexus complex when subjected to stretch. Forty-seven brachial plexus levels (identified by the four brachial plexus terminal nerve branches namely musculocutaneous, median, ulnar, and radial), obtained from 16 neonatal Yorkshire piglets (3–5 days old), were subjected to stretch-induced failure. The average maximum load and corresponding strain were reported to be 16.6 ± 1.3 N and 36.1 ± 1.6%, respectively. Maximum loads reported at the musculocutaneous level were significantly lower than the median and radial levels. No differences in strains at failure were reported at all four tested levels. Proximal or distal failure locations were reported in 83% of the tests with 17% mid-length ruptures that were primarily reported at the bifurcation of the median and ulnar brachial plexus levels. Histological studies reported an overall loss of wavy pattern of the nerve fibers, an increase in nerve spacing, fiber disruptions, and blood vessel ruptures in the stretched tissue. This *in vivo* piglet animal study offers insight into the NBPP mechanism by reporting biomechanical, injury location, and structural damage responses in neonatal brachial plexus when subjected to stretch.

## Introduction

The reported incidence of neonatal brachial plexus palsy (NBPP) continues to be 1.5 per 1000 live births despite significant technological advancement in the field of obstetrics [1]. NBPP results from over-stretching of the neonatal brachial plexus during complicated birthing scenarios, such as shoulder dystocia, and can significantly impact the infants' quality of life [2–9]. The current standard of care allows proper diagnosis only after the first two-three months of birth and while 70–90% of affected infants recover spontaneously, 20–30% of affected infants do suffer permanent deficits such as decreased strength, size, and girth of affected muscles, and limited range of motion [5,8,10]. The delay of NBPP prognosis is inevitable due to the age

**Data Availability Statement:** All relevant data are within the manuscript and its Supporting information files.

**Funding:** This project was supported by funding from the Eunice Kennedy Shriver National Institute

of Child Health and Human Development of the National Institutes of Health under Award Number R15HD093024, R01HD104910A, and NSF CAREER grant Award #1752513. The funders had no role in study design, data collection, and analysis, the decision to publish, or preparation of the manuscript.

**Competing interests:** The authors have declared that no competing interests exist.

of the infants, however, an understanding of the injury severity in these neonates can help predict the injury outcome. The lack of information on the biomechanical response of the neonatal brachial plexus complex when subjected to stretch limits our understanding of the NBPP injury mechanism and subsequently, its severity thereby limiting the outcome prediction.

Current available literature on the biomechanical properties of the brachial plexus complex includes *in vitro* studies in adult human cadaveric tissue [11–13] and in small adult animal models [14,15]. These studies have reported biomechanical responses of the brachial plexus complex by stretching the complex until failure at various loading rates and loading directions. Studies using adult human cadavers have reported an average failure force between 217–807 N, average failure stress between 1.3–3.5 MPa, and average failure strain between 19.6–58.8% [11–13]. Studies using small adult animals have reported an average failure force between 16–39 N, average failure stress between 6.7–7.0 MPa, and average failure strain between 22.9–25.1% [14,15]. Clearly, these currently available studies report a wide range of data for studied parameters that dictate the biomechanical properties of the brachial plexus complex. Moreover, these studies are performed in adults and not in neonates.

Due to ethical limitations, studies on neonatal human tissue are challenging [16,17]. Computational and physical models serve as good surrogates that are currently employed to understand and demonstrate the effects of applied forces on the neonatal brachial plexus complex during simulations of complicated birthing deliveries [16–18]. Nevertheless, these models are limited as well since the neonate brachial plexus biomechanical properties used in these models are that of an adult rabbit tibial nerve [19]. Because brachial plexus is an anatomically complex structure that originates as an extension from the ventral rami of C5 to Th1 (C: Cervical and Th: Thoracic) nerve roots, followed by trunks, divisions, cords, and terminal nerve branches, representing the biomechanical response of the neonate brachial plexus complex using the biomechanical response of a tibial nerve does not fully capture the brachial plexus complex response [5,9,20,21]. To date, only one study exists that reports the *in vitro* biomechanical properties of the neonatal brachial plexus complex when subjected to stretch using a neonatal large animal model [22]. Anatomical similarities exist between piglet and human neonate brachial plexus where the brachial plexus in both form an upper (i.e., C5-C6), middle (i.e., C7), and lower (i.e., C8-Th1) plexus [23]. Singh et al., 2018 study was the first to report varying biomechanical properties among individual brachial plexus segments and the tibial nerve of a piglet [22]. However, this study reported the *in vitro* biomechanical properties of individual brachial plexus segments and not the entire brachial plexus complex. Furthermore, there are no studies that have used a large neonatal animal model to perform *in vivo* biomechanical testing and report the brachial plexus responses. An *in vivo* study would account for plexus anatomy, surrounding connective tissue, and plexus environment and help enhance our understanding of the stretch injury mechanisms that occur in infants during complicated birthing scenarios.

This study aims to fill that critical gap by using a neonatal piglet animal model and provide an understanding of the *in vivo* tensile biomechanical properties of the neonatal brachial plexus complex when subjected to stretch.

## Methods

All surgical procedures were approved by the Institutional Animal Care and Use Committee (IACUC). Fig 1 outlines a summary of the methodology.

### Surgical preparation

A total of 16 neonatal Yorkshire piglets (3–5 days old) were used in this *in vivo* study. Piglets fasted for a minimum of 2 hours prior to surgical procedures and were then sedated by an

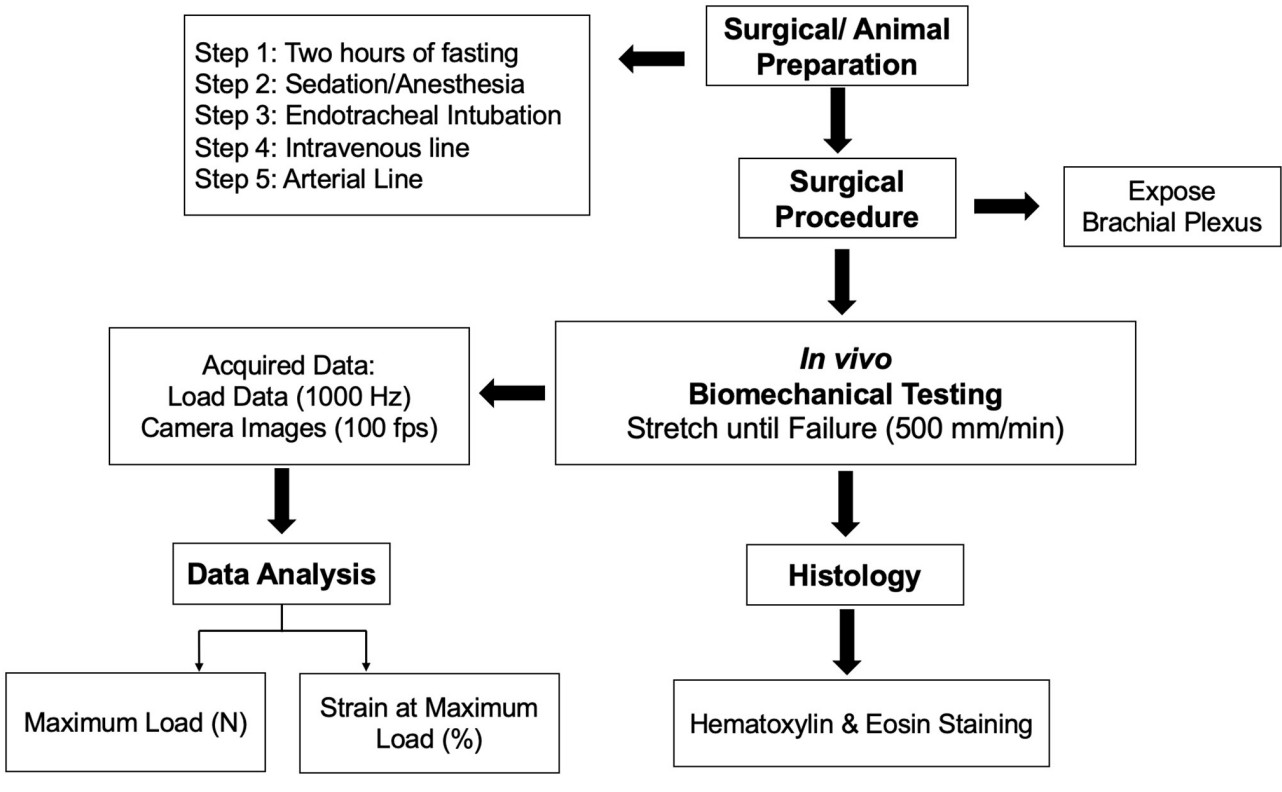

**Fig 1. Flowchart of research methodology.**

intramuscular injection of ketamine (20 mg/kg) and xylazine (1.5 mg/kg). Animals then received 4% isoflurane via mask until they reached a plane of surgical anesthesia, assessed by the absence of canthal reflex and withdrawal response to a toe pinch. Piglets were then endotracheally intubated and maintained on a 2% isoflurane mixed with 50:50 oxygen and nitrous oxide gases. The intravenous and arterial lines were established in the subcutaneous abdominal vein and femoral artery, respectively. Physiological monitoring included the placement of electrocardiogram surface electrode leads on the four extremities to measure heart rate, a pulse oximeter on the skin of the ear to measure heart rate and oxygen saturation, a rectal thermometer to measure core body temperature, and a sampling tube attached to the endotracheal tube to measure end-tidal carbon dioxide. Piglets were continuously monitored, and the physiological parameters were collected and recorded every 15 minutes. Arterial blood gases (pH, $pO_2$, $pCO_2$) were also evaluated every 30 minutes until the end of the experiment.

## Surgical procedures

Piglets were placed in a supine position with upper limbs in abduction to expose the axillary region. A midline incision was made through the skin and fascia overlying the trachea down to the upper third of the sternum corresponding to the spinal levels between the third cervical (C3) and thoracic (Th3) vertebrae. Using blunt dissection, the superior and inferior flaps were released to expose the cervical and thoracic segments of the entire brachial plexus complex, respectively. The brachial plexus complex was then carefully examined to locate the bifurcations of the divisions (M shape). brachial plexus segments were then identified relative to these

bifurcations such that segments closer to the spine were labeled as root/trunk and those below these bifurcations were labeled as cord/nerve segments.

### Test apparatus for *in vivo* biomechanical testing

A custom-made mechanical testing set-up described previously by Singh et al., 2019 [24] was used to induce mechanical stretches on the brachial plexus. The set-up consisted of an actuator, a load cell, a clamp, and a three-dimensional (3D) imaging system (Fig 2A). Briefly, an 8x8-inch aluminum base anchored to a portable cart (by two C-clamps, not shown) served as a portable base for the testing apparatus. A two-foot-long aluminum T-slot was threaded into the aluminum base. An actuator housing unit that could swivel was attached to the vertical end of the T-slot that housed a feedback linear actuator (Progressive Automations, Arlington, WA, USA). The actuator was connected to a sub-miniature 200 N load cell (Omega, Norwalk, CT, USA) that was then attached to a customized nerve clamp [25].

A stereo-imaging system (Fig 2A), as described previously, was utilized to track tissue displacement during mechanical testing. A ZED Mini camera (StereoLabs, San Francisco, CA, USA) was placed directly above the tissue being tested. The ZED Mini is a passive stereo camera with two horizontally aligned cameras separated by 63 mm (100 frames per second, FPS). To obtain the 3D points of the tissue displacement from the two (left and right) camera images, the direct linear transformation (DLT) method was used.

### *In vivo* biomechanical testing

In each animal, bilateral biomechanical testing was performed at three-four brachial plexus levels that were identified by the four brachial plexus terminal nerve branches (musculocutaneous, median, ulnar, and radial). These nerve branches were cut just prior to stretching at the distal end and anchored to the clamp of the testing apparatus (Fig 2B). Prior to stretch, using black acrylic paint, two to four markers were placed along the entire tissue length (over various segments) of the clamped brachial plexus level. The ZED Mini stereo-camera was positioned

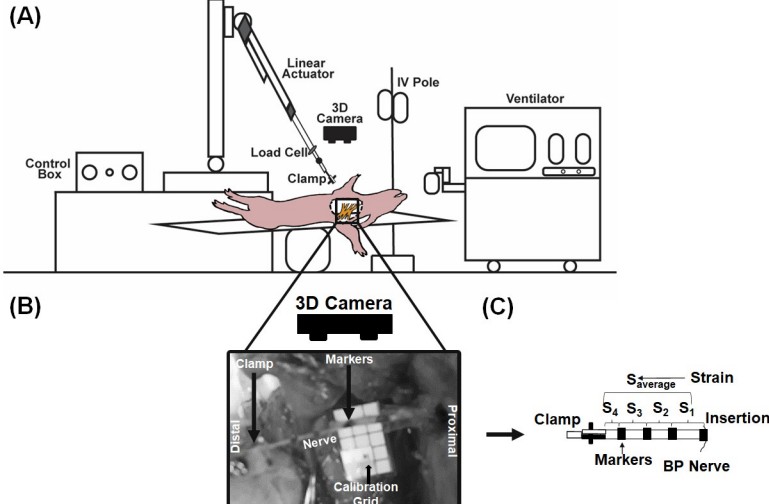

**Fig 2.** (A) *In vivo* biomechanical testing apparatus. The frame accommodates a computer-controlled actuator and a load cell. A portable stand (not shown) held the ZED stereo-imaging system. (B) Image of a brachial plexus level during testing. (C) Strain (S) was measured between adjacent makers ($S_1$, $S_2$, and so on) of the brachial plexus segments of the tested level. $S_{average}$ defines the average strain on the tested brachial plexus level.

above the clamped brachial plexus level to capture images of the markers placed on the tissue during the stretch. These images were later used to determine the *in-situ* tissue strain during the pull (Fig 2C). The actuator, load cell, and camera were then triggered synchronously using a custom MATLAB (MathWorks, Natick, MA, USA) code. The actuator was set to subject stretch on the brachial plexus segment at a 500 mm/min rate until complete failure occurred. The load and displacement data were acquired at 1000 Hz. Images were captured at 100 FPS. After each test, the clamp was checked for the presence of tissue. No tissue in the clamp implied the brachial plexus nerve segment slipped and the results of those experiments were discarded.

## Data analysis

For strain analysis, DLTdv Digitizing Tool [26], was used to track the displacements of the markers in the two-dimensional (2D) image plane of the left and right videos that were obtained from the left and right cameras, respectively. Using the DLT calibration coefficients, the 3D points (in the x, y, and z directions) were calculated from the tracked 2D image points obtained from the left ($u_l$, $v_l$) and right ($u_r$, $v_r$) videos. Using a custom MATLAB (MathWorks, Natick, MA, USA) code, the 3D points were imported and used to calculate the length (*l*) of the nerve (defined between the insertion and clamp) and the distance between each adjacent marker at each time point during stretch using Eq 1.

$$l_i = \sqrt{\left(x_{2_i} - x_{1_i}\right)^2 + \left(y_{2_i} - y_{1_i}\right)^2 + \left(z_{2_i} - z_{1_i}\right)^2} \tag{1}$$

Where $l_i$, is the distance between any two markers at any time point. $x_{1i}$, $y_{1i}$, $z_{1i}$ are the 3D coordinates of one of the two markers and $x_{2i}$, $y_{2i}$, $z_{2i}$ are the 3D coordinates of the second marker.

The change in length ($\Delta l$) between each pair of adjacent markers at each time point was calculated using Eq 2.

$$\Delta l_i = l_i - l_o \tag{2}$$

Where $l_i$, is the distance between any two markers at any time point and $l_o$ is the distance between any two markers at the original/zero time point.

Percent strain was then determined between insertion and clamp and between each adjacent marker at every time point using Eq 3.

$$percent\ strain_i = \frac{\Delta l_i}{l_o} * 100 \tag{3}$$

Where $\Delta_{li}$, is the change of length of any two markers at any time point.

The relationship between load, displacement, and image data could be characterized because they were recorded synchronously. Using these data, the load-time and strain-time curves were then plotted. Using these plots, the maximum load (identified as the point before gross rupture of the brachial plexus nerve) and strain at maximum load were determined. The camera data were also used to track changes in the structural integrity of the brachial plexus complex and to determine the failure location (proximal, mid-length, or distal).

## Histological studies

Two additional normal animals were included in the histological studies to provide unstretched tissue samples (n = 16 brachial plexus levels). At the end of the experiments, animals were euthanized (Euthasol), and the four brachial plexus levels were harvested and stored

in 4% paraformaldehyde in phosphate buffered saline (PBS) for 48 hours. After incubation with 30% sucrose/PBS for cryoprotection, longitudinal sections (10μm) were prepared on a cryostat for Hematoxylin and Eosin (H&E) staining. Sections were then studied by bright-field light microscopy for structural integrity at 10x magnifications. Images were acquired using a Leica DM4000 B microscope (Wetzlar, Germany) and saved using InFocus software (InFocus, Portland, OR, USA).

### Statistical analysis

Statistical analysis was performed using SPSS software, version 26 (SPSS Inc., Chicago, IL, USA). The maximum load and strain at maximum load are reported as mean ± standard error of mean (mean ± SEM). A non-parametric Kruskal Wallis test and Bonferroni post-hoc tests were performed to compare the maximum load, and strain at maximum load with one independent variable, namely brachial plexus levels (musculocutaneous, median, ulnar, and radial) with $p < 0.05$ considered significant. No quantification studies were performed for histological slides.

## Results

The *in vivo* failure biomechanical response of the neonatal brachial plexus was analyzed from 16 piglets. A total of 55 brachial plexus levels were stretched at a rate of 500 mm/min. However, due to slippage, 8 tests were discarded and not included in the data analysis. Table 1 summarizes the total number of brachial plexus levels (n = 47) included in the data analysis and the obtained *in vivo* biomechanical properties of the neonatal brachial plexus complex obtained from these failure tests. Representative load-time and strain-time (observed between the first and last markers) plots are shown in Fig 3A and 3B, respectively.

### Biomechanical responses of various brachial plexus levels

The *in vivo* failure biomechanical properties of the four brachial plexus levels identified by their terminal nerve branches, namely, musculocutaneous, median, ulnar, and radial are shown in Table 2. A significant difference was found in the maximum load sustained by the four stretched brachial plexus levels (Kruskal-Wallis test, $p < 0.05$); while no significant differences were found in the corresponding strain values between these levels ($p = 0.650$).

Pairwise comparisons between the four brachial plexus levels (musculocutaneous, median, ulnar, and radial) reported the maximum load of the musculocutaneous level to be significantly lower than median and radial levels ($p < 0.05$). No pairwise differences in the strain at maximum load were found between the four tested brachial plexus levels (p = 0.506) as shown in Fig 4.

### Failure location during stretch of various brachial plexus levels

Failure was noted along the entire length of the brachial plexus segments. Out of the 47 stretched brachial plexus nerve levels, ~40% (19/47) failed proximally, ~17% (8/47) failed mid-

**Table 1.** *In vivo* **maximum load [N] and strain at maximum load [%] (mean ± SEM) of the tested brachial plexus levels.**

| Sample Summary and Biomechanics Properties of Neonatal Brachial Plexus | |
|---|---|
| Rate | 500 mm/min |
| # of piglets | 16 |
| *n* (tested brachial plexus levels) | 47 |
| Maximum Load [N] | 16.6 ± 1.3 |
| Strain at Maximum Load [%] | 36.1 ± 1.6 |

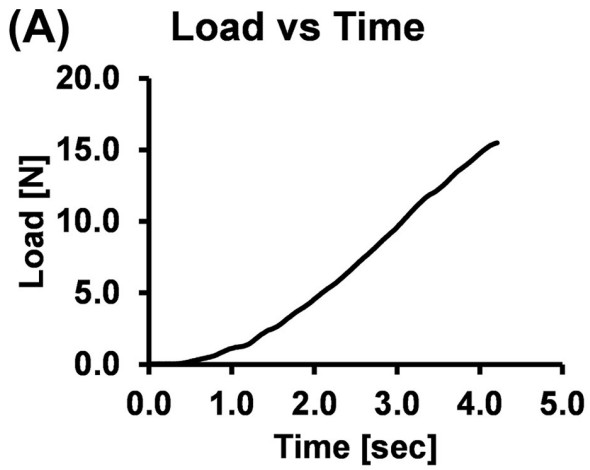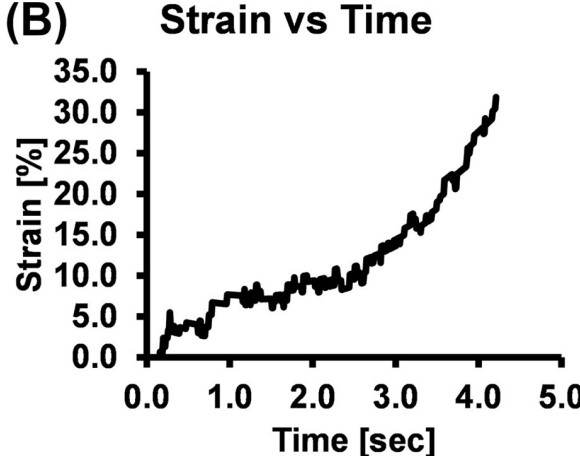

**Fig 3. Representative (A) load-time plot and (B) strain-time plot of the clamped brachial plexus level when subjected to failure stretch.**

length, and ~43% (20/47) failed distally ([Fig 5]). Proximal failure was defined as failure closer to the spine at the insertion site including the root segment, mid-length failure included failure along the length of the brachial plexus segments including the trunk, division, and cord segments, and distal failure was failure closer to the clamp including the nerve segment. Failure locations for each of the four tested brachial plexus levels are summarized in [Table 3].

### Histological studies

H&E-stained sections of stretched and unstretched (obtained from two control animals) brachial plexus complex levels were imaged and observed in brightfield at 10X. An overall loss of wavy pattern of the nerve fibers, increase in nerve spacing, fiber disruptions, and blood vessel ruptures was evident in stretched tissue when compared to unstretched tissue ([Fig 6]).

### Discussion

Neonatal (high birth weight >4000 grams), maternal (age >35 years, abnormal pelvic anatomy), and birth-related (delivery management, time, and mode of delivery) risk factors can lead to complicated delivery scenarios where an over-stretching of the neonatal brachial plexus can occur [2,3,5,7,20,27]. Studies have reported that the forces that the neonatal brachial plexus may experience are proportional to the magnitude, loading rate and direction, and surrounding connective tissue properties [6,17,28,29]. Currently, available literature on the biomechanical responses of the brachial plexus complex when stretched remains poorly studied. This

**Table 2. *In vivo* maximum load [N] and strain at maximum load [%] of brachial plexus levels, when subjected to failure, stretch at 500 mm/min.**

| Nerve Terminal (*n*) | Musculocutaneous (16) | Median (10) | Ulnar (8) | Radial (13) |
|---|---|---|---|---|
| **Maximum Load\* [N]** | 7.8 ± 0.5 | 19.6 ± 1.4 | 14.9 ± 0.7 | 26.9 ± 2.5 |
| **Strain at Maximum Load [%]** | 36.7 ± 2.6 | 34.3 ± 4.5 | 33.0 ± 1.1 | 38.7 ± 3.3 |

*Note*: Values shown in mean ± SEM.

\*Values were found significantly different between the four stretched brachial plexus levels.

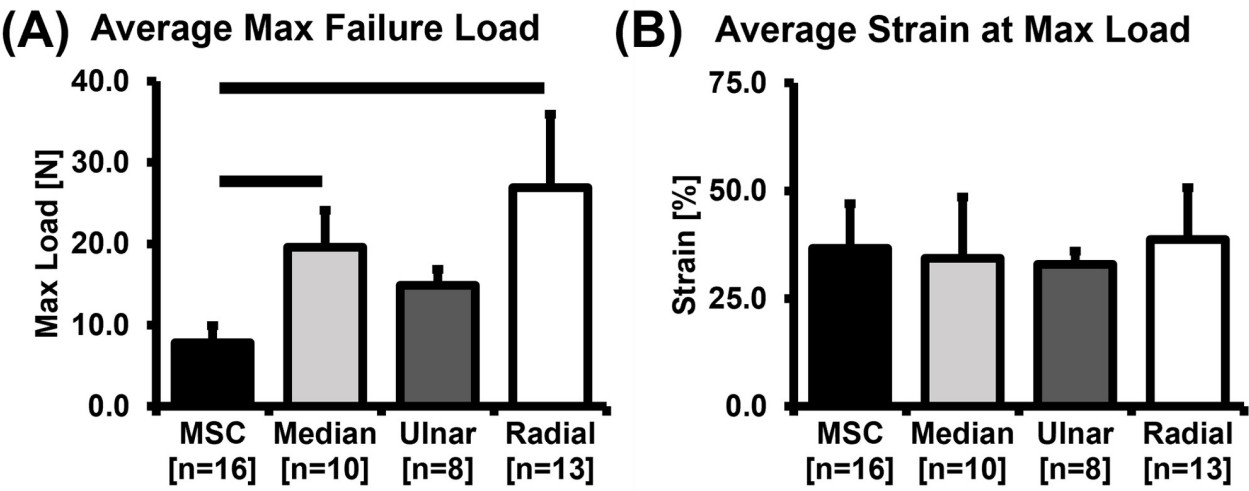

**Fig 4. Bar graphs of mean ± SD values of (A) maximum (max) load and (B) strain at maximum load of the brachial plexus levels identified by their terminal nerve branches (musculocutaneous, median, ulnar, and radial) when stretched to failure at 500 mm/min.** Error bars represent SD values. A dark solid line above the bars represents significant differences ($p < 0.05$) in the values between each brachial plexus level. MSC: Musculocutaneous.

discrepancy can be attributed to the variability in testing approaches of the brachial plexus tissue, tissue handling, species differences, and other factors. Moreover, most reported data on biomechanical responses of brachial plexus when subjected to stretch are based on adult human cadaveric tissue [11–13] or small adult animal tissue [14,15] due to ethical limitations in acquiring human neonatal brachial plexus tissue. Biomechanical studies that utilize a clinically relevant large animal model, such as the neonatal porcine can help overcome this

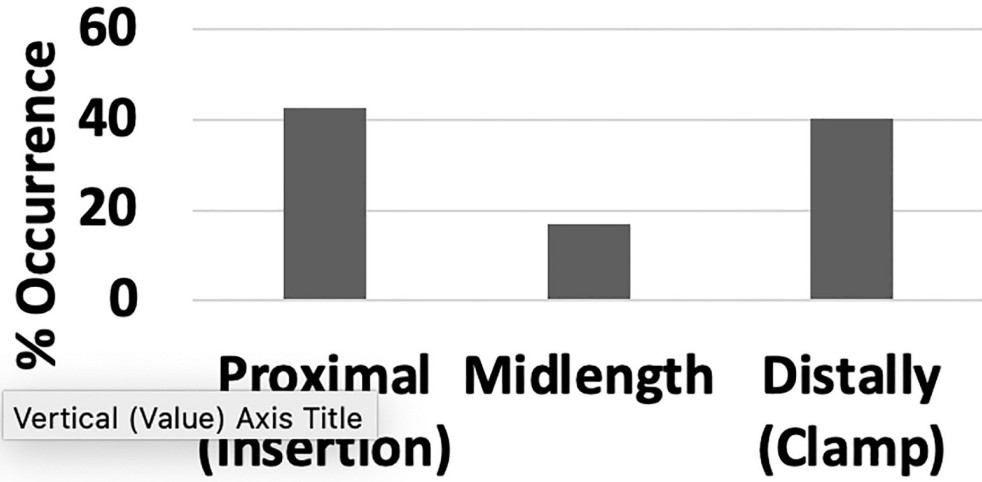

**Fig 5. Failure location summary for all tested brachial plexus levels.** Most failures were observed proximally and distally.

**Table 3. Details of failure locations for the four brachial plexus levels, when subjected to failure stretch at 500 mm/min.**

|  |  | Brachial Plexus Levels (n) | | | |
|---|---|---|---|---|---|
|  |  | Musculocutaneous (16) | Median (10) | Ulnar (8) | Radial (13) |
| *Failure Location* | Proximal | 56.25% | 30% | 25% | 46.20% |
|  | Mid-Length | 12.50% | 30% | 37.50% | 0.00% |
|  | Distal | 31.20% | 40% | 37.50% | 53.80% |

limitation [30,31]. The porcine nervous system has been shown to be comparable to that of the human nervous system, particularly with similarities in the size, length, and fascicular pattern of the brachial plexus segments [32]. Although piglets lack a clavicle and the piglet brachial plexus has less developed divisions between the three trunks and cords, the piglet and neonatal human brachial plexus both have an upper, middle, and lower brachial plexus defined by the C5-C6, C7, and C8-Th1 nerve roots, respectively [33]. Additionally, the piglet model has previously been used to study the extent of functional deficit of the brachial plexus in response to injury [34]. To date, only one study exists that utilizes the neonatal piglet animal model to study the brachial plexus failure response when subjected to stretch [22]. However, this study reports the *in vitro* biomechanical properties of individual brachial plexus segments rather than the entire brachial plexus within its native anatomical location (i.e., surrounding connective tissue remains intact). The current study reports the *in vivo* biomechanical properties of the entire brachial plexus using a neonatal piglet animal model.

Singh et al., 2018 tested individually excised brachial plexus segments (i.e., root/trunk, cord, terminal nerve branch) from freshly euthanized piglets at two distinct rates (i.e., 0.6 mm/min and 600 mm/min) [22]. At the 600 mm/min rate, the average failure load of the brachial plexus was reported to be $3.52 \pm 0.42$ N. In contrast, the current study reports the average failure load of the brachial plexus to be $16.6 \pm 1.3$ N at a 500 mm/min stretch rate. The observed higher failure load of the brachial plexus in the current study can be attributed to the difference in the testing environment between the two studies. Previous studies found peripheral nerves

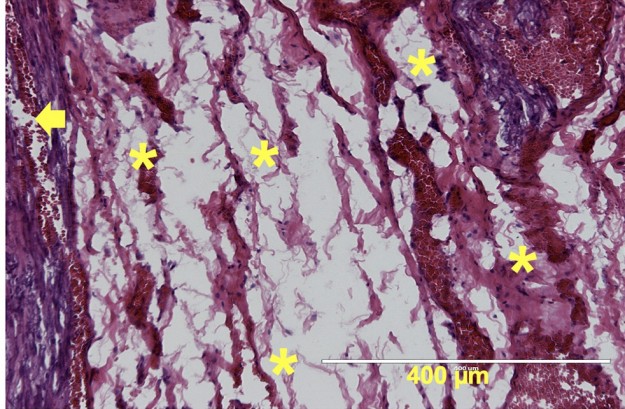

**Fig 6. H&E-stained sections of brachial plexus nerves (Ulnar).** Left (A): Normal brachial plexus Ulnar nerve segment. Right (B): Stretched (Failed) brachial plexus Ulnar nerve segment. 10x magnification. Torn fibers (*), broken blood vessels (arrowhead), and increased spacing are evident in the failed tissue.

stretched in their native anatomical location observed higher loads and stiffness, suggesting the surrounding connective tissue affects tensile mechanical properties [35,36]. The current study performed *in vivo* stretch of the various brachial plexus levels (i.e., identified by their terminal nerve branches) within its native anatomical location of anesthetized piglets, while Singh et al., 2018 performed *in vitro* stretch of excised isolated brachial plexus segments from freshly euthanized piglets [22]. Thus, the current study reports the *in vivo* failure response of the entire brachial plexus and not the *in vitro* failure response of isolated brachial plexus segments.

In another study, Kawai et al., 1989 performed stretch studies on the entire brachial plexus in an adult rabbit animal model at three distinct loading directions (i.e., upward, lateral, downward) at a rate of 500 mm/min [14]. It was found that loading direction was directly associated with the failure site and level. The reported failure loads ranged between 20 to 38 N of the entire brachial plexus when stretched at distinct brachial plexus directions. At the same displacement rate, the current study reports a lower failure load for brachial plexus levels, which is 16.6 ± 1.3 N. Differences in the testing approach between the two studies can contribute to the observed lower failure load in the current study. Kawai et al., 1989 stretched the entire brachial plexus complex, while the current study subject stretched only one brachial plexus level that was identified by its terminal nerve branch. Another difference between the two studies includes different animal species and ages that were used. Kawai et al., 1989 used a small adult animal model (i.e., rabbit) while the current study used a neonatal large animal model (i.e., piglet). Variations among species may affect brachial plexus biomechanical properties because of differences in brachial plexus tissue size and fiber pattern, as well as nerve roots defining the brachial plexus [32,37]. A previous study reported rabbit brachial plexus extends from nerve roots C5, C6, C7, C8, Th1, and Th2 (sometimes) and has an upper and lower trunk [37], which differs from both the piglet and neonatal human brachial plexus that have an upper, middle, and lower brachial plexus trunk [33].

In a series of human cadaveric studies, Zapalowicz stretched the entire adult human cadaveric brachial plexus complex at varying loading directions [11–13]. In their 2000 study, the failure load ranged between 217.7–546.3 N when stretched at a rate of 10 mm/min with a lateral loading direction [11], and in their 2005 and 2018 studies, the failure load ranged between 365 to 807 N when stretched at a rate of 200 mm/min in either parallel, perpendicular, or 45˚ caudal loading directions [12,13]. For these studies, the entire brachial plexus complexes were harvested from adult human cadavers that had no history of neurological disease. The sample preparation included isolating the entire brachial plexus from surrounding muscles and connective tissues while the brachial plexus terminal nerve branches were cut at the axilla. The spinal column was then secured at the static end while all brachial plexus terminal nerve branches were clamped at the moving end of the testing apparatus [11–13]. In contrast to their approach, the current study, only stretched one brachial plexus level while preserving the brachial plexus native anatomical surrounding. Additionally, the current study reports the *in vivo* response of neonatal brachial plexus using a piglet animal model, while Zapalowicz reports the *in vitro* response of adult brachial plexus using adult human cadaveric tissue. Knowing that there are age-dependent differences in peripheral nerves, such that the ultimate stress of adult (ages 20 to 69 years old, $1.28 \pm 0.016$ kg/mm$^2$) sciatic nerves have been reported to be greater than adolescents (ages 0 to 19 years old, $1.14 \pm 0.035$ kg/mm$^2$) and neonates (age one-month-old, $0.96 \pm 0.026$ kg/mm$^2$) sciatic nerves [38], findings from the current study offer a better understanding of NBPP injury mechanisms.

Severe cases of brachial plexus injury can include avulsion or rupture injuries. In the current study, we recorded the failure location along the brachial plexus segment for the tested

levels to better understand the failure mechanisms among the various brachial plexus levels. In the current study, three failure locations were defined, namely proximal (avulsion/rupture), mid-length (rupture), or distal (rupture closer to the clamped end). At the musculocutaneous brachial plexus level, which exits from C5-C6 spinal levels, proximal failures (avulsion/rupture) were more common and occurred in 56.2% of the cases, followed by distal failures (31%) and a few mid-length (12%) failures. This is relevant to the clinically reported Erb's palsy (root injuries at C5-C6) [39]. The radial level (axially aligned to the C7 exit) also reported failure occurrences at either the proximal (46%) or distal locations (54%) with none along the mid-length. In contrast, the medial and ulnar brachial plexus levels (exiting from C5-C6 and C8-T1 spinal levels) reported the highest failure occurrences at mid-length (median-30% and ulnar-37%) when compared to musculocutaneous (12%) and radial (0%) levels. The prevalence of the observed failure locations at the four tested brachial plexus levels in the current study can directly be related to the brachial plexus anatomy (Fig 6). Axial stretching of the musculocutaneous brachial plexus level directly transmits forces to the root/trunk segments of the brachial plexus that exit from C5-C6 levels, which offers mechanical justification for the observed 56% proximal failures. The radial brachial plexus level is thickest at the middle trunk and posterior cord thereby limiting mid-length rupture, which is reflected in the absence of any reported mid-length rupture at this brachial plexus level. The median and ulnar nerves transmit the axial stretches to the M-region bifurcations when stretched from the distal end. Mechanical considerations imply the M-region bifurcation to have the highest stress concentration when subjected to axial stretch, thereby supporting the high occurrences of mid-length rupture in these brachial plexus levels in the current study. Overall, the relationship between observed failure locations for various brachial plexus levels studied in the current study and its correlation to brachial plexus complex anatomy further emphasizes the effects of head and neck orientation on the direction of the line of action of force when understanding the injury mechanism of NBPP and investigating preventative strategies (Fig 7).

We also investigated the structural integrity of failed brachial plexus segments. Nerve segment data is presented in the current study since distal failure location was prevalent in all the tested brachial plexus levels. Previous studies have reported neurofilament disruption, increased spacing, blood vessel rupture, and impaired axoplasmic transport in stretched nerve tissue [25,40–42]. Using H&E staining, we observed increased spacing, broken blood vessels, and torn nerve fibers along the entire length of the nerve segment. We did not quantify the extent of damage; however, the images were observed carefully, and the extent of structural damage was similar among the various four tested brachial plexus levels.

In summary, this is the first study to report *in vivo* biomechanical properties at various levels of neonatal brachial plexus when subjected to stretch using a neonatal large animal piglet animal model. The study also offers insight into the failure location and an overview of associated structural changes in the brachial plexus nerve tissue when stretched. The biomechanical, injury location, and structural damage data offer insight into the injury mechanism responsible for NBPP. Furthermore, the obtained biomechanical parameters of the neonatal brachial plexus complex can be used to define detailed segmental-level biomechanical properties of the brachial plexus complex of the currently available neonatal computational model, thereby enhancing the biofidelic responses of the model. Utilizing this improved computational model, simulation outcomes can provide a better understanding of the currently employed clinical maneuvers and their effects on brachial plexus stretch. Furthermore, these simulation outcomes can guide delivery strategies that minimize the occurrence of NBPP during complicated birthing scenarios and can be used in the training of clinicians to advance the science of obstetrics care [43–46].

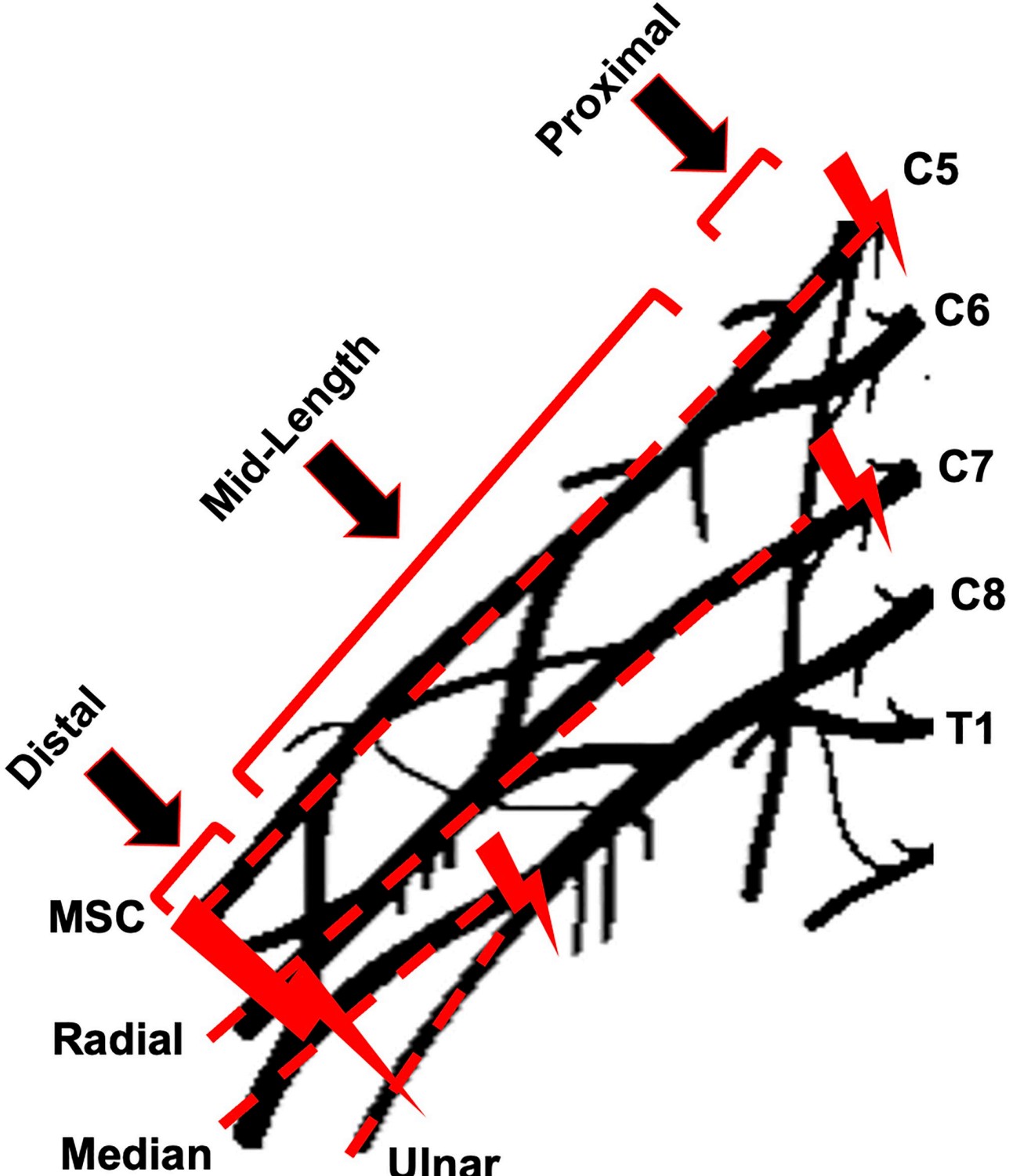

**Fig 7. Line of action of force (red dotted line) and failure location (red streak) in various tested brachial plexus levels.** Failure locations were identified as proximal (near insertion, root brachial plexus segment), mid-length (trunk/cord/division brachial plexus segments), and distal (near clamp, nerve brachial plexus segment). MSC: Musculocutaneous.

## Supporting information

**S1 File.**
(PDF)

## Acknowledgments

We would like to acknowledge Late Dr. Maria Delivoria for her guidance. We also thank Rachel Magee and Megan Gorleski for their assistance with the study.

## Author Contributions

**Conceptualization:** Anita Singh, Sriram Balasubramanian.

**Data curation:** Anita Singh.

**Formal analysis:** Anita Singh, Virginia Orozco.

**Funding acquisition:** Anita Singh, Sriram Balasubramanian.

**Investigation:** Anita Singh, Sriram Balasubramanian.

**Methodology:** Anita Singh, Virginia Orozco, Sriram Balasubramanian.

**Project administration:** Anita Singh, Sriram Balasubramanian.

**Resources:** Anita Singh, Sriram Balasubramanian.

**Software:** Anita Singh, Sriram Balasubramanian.

**Supervision:** Anita Singh, Sriram Balasubramanian.

**Validation:** Anita Singh.

**Visualization:** Anita Singh.

**Writing – original draft:** Anita Singh, Sriram Balasubramanian.

**Writing – review & editing:** Anita Singh, Virginia Orozco, Sriram Balasubramanian.

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
