## [Decision Letter · Decision Letter 0]

5 Jul 2023

PONE-D-23-14332In vivo biomechanical responses of neonatal brachial plexus when subjected to stretchPLOS ONE

Dear Dr. Singh,

Thank you for submitting your manuscript to PLOS ONE. After careful consideration, we feel that it has merit but does not fully meet PLOS ONE’s publication criteria as it currently stands. Therefore, we invite you to submit a revised version of the manuscript that addresses the points raised during the review process.

We look forward to receiving your revised manuscript.

Kind regards,

Richa Gupta

Academic Editor

PLOS ONE

Journal Requirements:

   "This project was supported by funding from the Eunice Kennedy Shriver National Institute of Child Health and Human Development of the National Institutes of Health under Award Number R15HD093024, R01HD104910A, and NSF CAREER grant Award #1752513"

   "This project was supported by funding from the Eunice Kennedy Shriver National Institute of Child Health and Human Development of the National Institutes of Health under Award Number R15HD093024, R01HD104910A, and NSF CAREER grant Award #1752513."

   "This project was supported by funding from the Eunice Kennedy Shriver National Institute of Child Health and Human Development of the National Institutes of Health under Award Number R15HD093024, R01HD104910A, and NSF CAREER grant Award #1752513." 

Additional Editor Comments:

Overall, this study is clinically quite relevant and innovative. This study is unique because most of the earlier studies have been conducted on adult cadavers, while this study has been conducted on neonatal animal piglet models.

Material & methodology – is well explained

Results - The results of this study can be quite helpful to design prevention strategies so as to minimize the incidence of neonatal brachial plexus injuries.

Discussion – Authors have described an association between the failure mechanisms among the various BP levels to different kinds of brachial plexus injury and its relation with brachial plexus anatomy. But still there are some interesting queries raised by reviewers, which need to be looked into. I am attaching the reviewer’s comments  which have been suggested. Authors are required to make the required amendments in their study, before we proceed with publication.

References are as per the submission guidelines

Reviewers' comments:

Reviewer's Responses to Questions

**Comments to the Author**

1. Is the manuscript technically sound, and do the data support the conclusions?

Reviewer #1: Yes

Reviewer #2: Yes

2. Has the statistical analysis been performed appropriately and rigorously? 

Reviewer #1: Yes

Reviewer #2: Yes

3. Have the authors made all data underlying the findings in their manuscript fully available?

Reviewer #1: Yes

Reviewer #2: Yes

4. Is the manuscript presented in an intelligible fashion and written in standard English?

Reviewer #1: Yes

Reviewer #2: Yes

5. Review Comments to the Author

Reviewer #1: very concise and good study. The study is very clinical relevant . there are minor grammatical errors but those are not relevant. Having piglets for the study presents with less discrepancy as compared to with humans

Reviewer #2: Research study is very unique and conducted keenly and excellently. Material and methods are explained in detail

Results are well explained

Statistical analysis done correctly.

I strongly recommend this article

6. PLOS authors have the option to publish the peer review history of their article (what does this mean?). If published, this will include your full peer review and any attached files.

Reviewer #1: No

Reviewer #2: **Yes: **Dr. Jasmeen Vajir Shaikh

---

## [Author Response · Author response to Decision Letter 0]

4 Aug 2023

Response to Editorial Comment

Comment: “Overall, this study is clinically quite relevant and innovative. This study is unique because most of the earlier studies have been conducted on adult cadavers, while this study has been conducted on neonatal animal piglet models. Material & methodology – is well explained Results - The results of this study can be quite helpful to design prevention strategies so as to minimize the incidence of neonatal brachial plexus injuries. Discussion – Authors have described an association between the failure mechanisms among the various BP levels to different kinds of brachial plexus injury and its relation with brachial plexus anatomy.

But still there are some interesting queries raised by reviewers, which need to be looked into. I am attaching the reviewer’s comments which have been suggested. Authors are required to make the required amendments in their study, before we proceed with publication.”

Response: We thank the Editor for their positive feedback and comments. We have addressed all queries raised by reviewers and made the required changes throughout the manuscript. Details of changes made are described in our response to the reviewer’s comments below. These changes are also highlighted in yellow throughout the manuscript.

Response to Reviewer Comments:

Reviewer 1:

Comment 1: Very concise and good study. The study is clinically very relevant. There are minor grammatical errors but those are not relevant. Having piglets for the study presents with less discrepancy as compared to with humans. – Accept.

Response: We thank Reviewer 1 for their positive feedback and their comments. We have addressed the minor grammatical errors throughout the manuscript. 

Reviewer 2:

Comment 1: Research study is very unique and conducted keenly and excellently. Material and methods are explained in detail. Results are well explained. Statistical analysis done correctly. I strongly recommend this article.

Response: We thank Reviewer 2 for their acknowledgment of our work and their comments. 

Reviewer 3:

Comment 1: Strong points – Novel study, in-vivo biomechanical properties of neonatal brachial plexus was studied. Neonatal piglet animal model was chosen, very much similar to human brachial plexus. Very extensive technology has been used to study bilateral brachial plexus. Extensive set up with an actuator, a load cell, a clamp & 3-D imaging system has been used to study the biomechanical properties at 3-4 points in BP. Time & effort has been put up in the study to locate the various failure points along the entire length of brachial plexus. Maximum load was calculated & also strain at maximum load was also noted. Histology of stretched tissue was performed to show changes in the tissue. Appropriate statistical analysis has been done. Graphs has been made to show the results obtained. May have clinical correlation in decreasing the overall incidence of stretching of NBPP. Weak points – Very stringent conditions are required to perform such extensive studies. This type of study requires a longer period of time. – Accept.

Response: We thank Reviewer 3 for their acknowledgment of our work and their comments. 

Comment 2: Authors can add clinical correlation of various parameters measured in the discussion section.

Response: We thank the reviewer for this suggestion and have now included the required in the Discussion Section of the revised manuscript (Lines # 382-388). Here we elaborate on how the obtained biomechanical parameters of the neonatal brachial plexus complex can be used to define detailed segmental-level biomechanical properties of the brachial plexus complex of the neonatal computational model, thereby enhancing the biofidelic responses of the model. Utilizing this improved model, simulation outcomes can provide a better understanding of the clinical maneuvers and their effects on brachial plexus stretch. Furthermore, these simulation outcomes can guide delivery strategies that minimize the occurrence of NBPP during complicated birthing scenarios.

Comment 3: One flowchart depicting various steps in the methodology section.

Response: We thank the reviewer for this suggestion. We have included a new figure (Fig. 1) that details a flowchart depicting the various steps in the methodology section. 

Comment 4: Authors can explain clinical implication of measuring maximum load & biomechanical properties of different nerves. 

Response: We thank the reviewer for reiterating the clinical implications of measured biomechanical parameters of various nerves in this study. We have included the required in the Discussion Section of the revised manuscript (Page #, Line #). 

Reviewer 4:

Comment 1: BP and MSC is not standard abbreviation so avoid using it.

Response: We acknowledge the reviewer’s suggestion of avoiding the use of BP and MSC as abbreviations of brachial plexus and musculocutaneous, respectively. We agree BP and MSC are not standard abbreviations and have accepted the recommendation to change BP to brachial plexus and MSC to musculocutaneous throughout the manuscript.

Comment 2: Line no 69 to 71 Only one study quoted which is by the same author which does not rule out bias. Please explain.

Response: We thank the reviewer for this comment. To address their concern, we want to mention that an extensive literature review was performed (including years 1955-2023) and only one study, performed by our group, was found to have reported in vitro biomechanical responses of neonatal brachial plexus in a large animal model such as a piglet. Building upon our previous work, our team has developed methodologies to perform similar experiments in vivo. Our goal here is to report our findings that will help advance our understanding of the biomechanical properties of neonatal brachial plexus tissue. While we understand the reviewer’s concern about the possibility of being biased, we wish to clarify that this study is primarily reporting our findings. We are not establishing any correlation with our previously reported work. We also clarify the use of the piglet animal model as a good surrogate due to anatomical similarities in the brachial plexus anatomy. We have included the justification for the use of the piglet model in the Introduction Section (Lines # 73-75).

Comment 3: Line 227 correct the spelling from chord to cord.

Response: We thank the reviewer for pointing out this mistake. This typo has been corrected in the revised manuscript.

Comment 4: Line 266 again study by the same author is quoted, need to justify.

Response: We agree with the reviewer and have addressed this concern in Comment 2. Additionally, we have also now included limitations of our previously reported study and how this study helps fill the critical gap. Introduction Section (Lines # 77-78) 

Comment 5: Line 277 the logic seems correct that surrounding tissue may allow to have much load stress but need to prove further.

Response: We acknowledge the reviewer’s suggestion of proving surrounding tissue may allow to have much load stress. We have described this further (Lines # 290-294) and added two additional studies (Millesi et al., 1990 and Ma et al., 2013) that compared in vivo and in vitro tensile mechanical properties of ulnar and median nerves. These two studies found that stretching peripheral nerves within their native anatomical location observed higher loads and greater stiffness. 

Comment 6: Did author study sensory and motor nerve threshold with stretch and load, If so is there any difference in sensory and motor load stress?

Response: We thank the reviewer for asking if sensory and motor nerve threshold with stretch and load was studied and if any difference was noted. The main objective of this current study was to report in vivo biomechanical properties of the neonatal brachial plexus subjected to stretch using a neonatal large animal model (i.e., piglet). No effort was made to study the individual threshold related to sensory and motor nerve. We appreciate the suggestion and will seek to investigate it in our future studies. 

Comment 7: Clinical utility of the study needs to be emphasized. Especially how to imply the finding of a study in routine practice to reduce the incident of NBPP.

Response: We agree with the reviewer and have included the clinical utility of the study in the Discussion section. Lines # 381-387 we elaborate on how the findings from this study can help enhance the biofidelic responses of the computational model that are the most promising surrogate while investigating clinical maneuvers that can help minimize the occurrence of NBPP during complicated birthing scenarios. 

Comment 8: Although a very good article suggest that even after the increase in the Cesarean section, the reduction in NBPP is not significant. Vaginal delivery may not be the only cause of it. (Semin Perinatol 2006 Oct;30(5):276-87. doi: 10.1053/j.semperi.2006.07.009.)

Cesarean section on request at 39 weeks: impact on shoulder dystocia, fetal trauma, neonatal encephalopathy, and intrauterine fetal demise Gary D V Hankins 1, Shannon M Clark, Mary B Munn – Accepted after Major Revision.

Response: We completely agree with the reviewer’s comment. Our goal is to understand the biomechanical response of neonatal brachial plexus, which is poorly understood currently, and then utilize this data to understand how NBPP is induced during birthing. Shoulder Dystocia is one of the associated factors for NBPP, but the data obtained is not limited to vaginal delivery scenarios. Future studies can explore the implication of this data in deliveries performed using Cesarean section. We are excited about this possibility and would consider investigating it as well in the future.

---

## [Editor Report · Decision Letter 1]

13 Aug 2023

In vivo biomechanical responses of neonatal brachial plexus when subjected to stretch

PONE-D-23-14332R1

Dear Dr. Singh,

We’re pleased to inform you that your manuscript has been judged scientifically suitable for publication and will be formally accepted for publication once it meets all outstanding technical requirements.

Kind regards,

Richa Gupta

Academic Editor

PLOS ONE

Additional Editor Comments (optional):

Authors have thoroughly addressed all the queries raised by reviewers and have made the desired modifications as suggested by the reviewers.

I would like to congratulate the authors for the meticulous work and very innovative study on biomechanical response of neonatal brachial plexus

We are happy to inform that article is ACCEPTED FOR PUBLICATION in Plos One
---

## [Editor Report · Acceptance letter]

18 Aug 2023

PONE-D-23-14332R1 

*In vivo* biomechanical responses of neonatal brachial plexus when subjected to stretch 

Dear Dr. Singh:

I'm pleased to inform you that your manuscript has been deemed suitable for publication in PLOS ONE. Congratulations! Your manuscript is now with our production department. 

Kind regards, 

on behalf of

Dr. Richa Gupta 

Academic Editor

PLOS ONE